# Research on the Influence of Risk on Construction Project Performance: A Systematic Review

**Guiliang Su [1],* and Rana Khallaf [2]**

1   Smart Construction and Innovation Center, School of Civil and Architectural Engineering, Guilin University of Technology, Guilin 541000, China
2   Department of Structural Engineering and Construction Management, Future University in Egypt, New Cairo 11835, Egypt; rana.khallaf@fue.edu.eg
*   Correspondence: guiliang_2000@glut.edu.cn

**Abstract:** Knowledge on the influence of risk on project performance is an important part of risk management. Previous studies have concentrated on this area to identify risks in aspects of project performance such as cost and schedule. However, rare reviews have been conducted to fully report on the research on the influence of risk on project performance. For this reason, to identify and analyze such researches in these areas a systematic review was conducted in this paper. More specifically, 54 relevant articles were identified and classified into three groups according to their research contents; the research contents in each article and the research methods were reviewed, and the 13 most frequent research methods were also identified and discussed. It was found that most of the previous researches concentrated on developing tools or approaches to assess the influence of risk on project performance. Additionally, researches focused on risk factors' identification or risk interdependency modeling were also common, along with researches that investigated the cause of poor project performance, evaluated risk impact on cost contingency, discussed the risk response actions, and discussed what enables high-risk projects to yield a high return. However, four gaps were identified from these researches, namely: a need for improving the accuracy in quantitative research of the influence of risk on project performance; a need for novel research methodologies for conducting more accurate risk influence assessments; taking into consideration project participants' decision-making in their researches; and creating a framework that treats the risk influence assessment as a whole system. Besides that, since this research only focused on two project objectives (cost and schedule), recommendations for future research include expanding the focus to more project objectives.

**Keywords:** risk influence; construction project performance; contents analysis; method analysis; review

## 1. Introduction

In recent years, there has been both an increase in the complexity of the projects delivered and an improvement in the techniques and technologies available, which has led to a focus on research related to risk management [1,2]. Throughout this trend, one of the most important focus points has been on researching the influence of risk on project performance. Previous research has been conducted in several areas, including developing tools or approaches to model the cost escalation uncertainty [3]; investigating the effect of risks on project schedule performance [4] and assessing the impacts of risk occurrence on project performance [5]; identifying elements or paths that have an influence on project performance [6,7]; and investigating the major causes of poor time and cost performance [8]. However, very few reviews have been conducted to fully report on the research on the influence of risk on project performance. Besides that, obtaining knowledge in this area would aid project participants in understanding the mechanism of risk impacts [9,10], conducting more accurate project performance estimations [11], creating robust risk registries

for projects [12], and selecting suitable risk response actions [13]. Therefore, this paper focuses on identifying and reporting on the influence of risk on project performance in order to highlight the research status and trends in this domain.

According to the Project Management Institute (PMI), risk is defined as an uncertain event that can have a positive or negative effect on the project objectives, which are time, cost, scope, and quality [14,15]. For the purpose of this research, the influence of risk is defined as a risk's positive or negative effect on the project objectives. According to Assaad et al. (2020), in these objectives, cost and schedule overruns are always regarded as being of paramount importance in the project control area [16]. Additionally, Johnson and Babu (2020) concentrated on analyzing the major causes of this poor time and cost performance in the UAE construction industry [8], while Dikmen et al. (2012) used an integrated duration–cost influence network model to analyze the influence of risk on project performance [17]. Therefore, this paper will focus on the effects of risk on cost and schedule in order to improve the project controls process.

Furthermore, previous researches have studied the influence of risk in certain scenarios. According to these researches, there are three risk relation types in construction projects: risk decision relations, risk influence relations, and risk interaction relations [18]. Risk decision relations refer to project participants' decision-making in risk situations [8], risk influence relations refer to the influence relations between risk and project objectives [19,20], and risk interaction relations refer to the interdependency of risk factors [6,9]. Additionally, the expected project performance during the project implementation process can also affect participants' decision-making in certain areas, especially in some project types (such as IPD or IPD-like projects), for the adoption of "shared risk and reward" principles in their incentive mechanism [18,21]. Therefore, research on the influence of risk on project performance should be considered as an interaction process, as shown in Figure 1. That is, it can be classified into three groups, as follows: (a) influence of risk factors and risk interdependency on project performance; (b) influence of risk factors and risk interdependency on project performance while taking into consideration project participants' simultaneous decision-making on risk; (c) treat the interaction process as a whole (as shown in Figure 1) and study the influence of risk on project performance while considering the feedback loop. The difference between (b) and (c) is that (c) takes into consideration the feedback effect between project performance and stakeholders' behaviors (decisions), which could influence stakeholders' decisions and then influence the project risk and the project performance. However, (b) does not take into consideration the feedback effect between project performance and stakeholder behaviors. This becomes especially difficult with the increased number of contracting parties in a project [22].

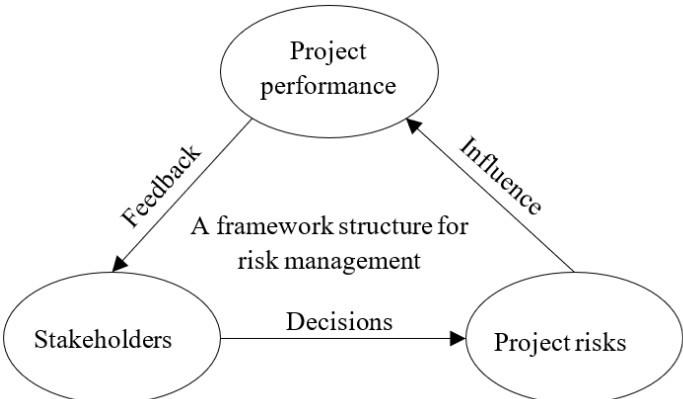

**Figure 1.** Framework for risk management [18].

Previously, several reviews have been conducted on risk management. Khallaf et al. (2018) proposed a three-tier classification of risks in public-private partnership projects [12]; Taroun et al. (2011) and Taroun (2014) reviewed construction risk modeling and assess-

ment [23,24]; Zou et al. (2017) concentrated on the review of risk management through BIM and BIM-related technologies [25]; Xia et al. (2018) conducted a review on the integrations of construction risk management and stakeholder management [26]; Siraj et al. (2019) reviewed the risk identification and common risks in construction [27]; Hegde and Rokseth (2020) carried out a review on the applications of machine learning methods in engineering risk assessment [28]. Many of these researches have focused either on risk identification or risk modeling and assessment. However, there have been no previous reviews targeting the four groups, as mentioned earlier. For this reason, a systematic review is needed on the influence of risk on project performance.

The paper is organized as follows: firstly, the research background is introduced; secondly, the research methodology section, including the methodology process and a discussion on the inclusion criteria and the identified articles, is provided. thirdly, a review of research contents and research methods in the articles is presented; and finally, a discussion of the identified articles is presented, along with the conclusion.

## 2. Methodology

### 2.1. Methodology Process

The research method adopted in this paper consists of five steps, as shown in Figure 2. This follows the systematic literature review methodology, which has been used in the literature to report on a specific topic [12,29–31]. In Step 1, the inclusion and exclusion criteria and the articles to be targeted for analysis are identified through a systematic search process. Statistics are also created to explore the general characteristics of the identified articles. In Step 2, the main research contents of the articles are identified and analyzed. In Step 3, a systematic investigation is conducted on the identified articles, the research methods used, and their applications. In Step 4, a systematic discussion on the findings is presented, and, finally, recommendations for future work are made in Step 5.

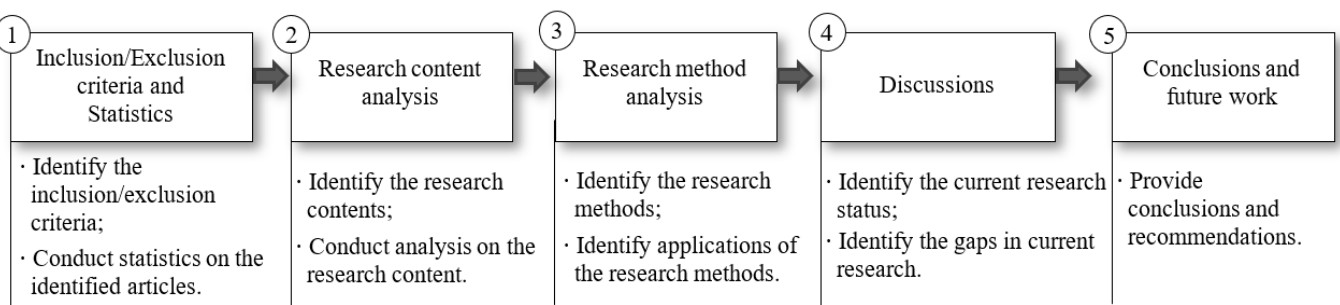

**Figure 2.** Research flowchart.

### 2.2. Inclusion/Exclusion Criteria and Statistics

2.2.1. Identification of the Inclusion and Exclusion Criteria

a.    Journal Selection

The first step in identifying the articles is journal selection. According to Siraj et al. (2019), for a review on project risk management, the journals that have a CiteScore of 0.70 and above and also contain articles specifically on risk-related topics by different authors should be considered [27]. In this paper, the authors followed Siraj et al. (2019)'s method for journal selection. That is, the selected journals should have a CiteScore of 0.70 and above and contain more than three articles relevant to the topic. Hence, the 14 journals identified by Siraj et al. (2019) were used to conduct the article survey [27]: *Expert Systems with Applications* (ESA), *Automation in Construction* (AC), *International Journal of Project Management* (IJPM), *Building and Environment* (B&E), *Journal of Construction Engineering and Management* (JCEM), *Journal of Computing in Civil Engineering* (JCCE), *Journal of Management in Engineering* (JME), *Journal of Infrastructure Systems* (JIS), *Construction Management and Economics* (CME), *Journal of Civil Engineering and Management* (JCiEM), *Engineering, Con-*

*struction and Architectural Management* (ECAM), *Canadian Journal of Civil Engineering* (CJCE), *International Journal of Construction Management* (IJCM), and *ASCE-ASME Journal of Risk and Uncertainty in Engineering Systems, Part A:Civil Engineering* (JRUES). Additionally, another four journals with a large number of publications that are related to the topic in this paper and have a CiteScore of 0.70 and above were also surveyed: *Journal of cleaner production* (CiteScore = 13.1), *Sustainability* (CiteScore = 3.9), *Risk analysis* (CiteScore = 6.0), and *Safety science* (CiteScore = 7.8).

b.　　Article Selection

In this paper, a systematic review process was used to report on the influence of risks on project performance. In order to conduct a systematic review, the first step is identifying keywords to be used for conducting the search. The keywords used in this research were: "risk" AND "construction" along with "assessment" OR "performance" to search the Scopus and Web of Science databases. Secondly, the time period for the search was limited to between 2001 and 2021. In these steps, sources including journals, proceedings, books, etc., in Scopus or presented in Web of Science were retained and surveyed.

Finally, as shown in Figure 3, the resulting papers were systematically screened and identified. In the beginning, 3405 records were identified from the Scopus and Web of Science databases. The records that were outside the scope or did not address the influence of risk on cost performance or schedule performance in construction projects were deemed irrelevant records and removed. The next step was conducting a review of the abstract for the remaining articles to finalize the list. The result was 54 papers in 14 journals, which were shortlisted for the review process. The detailed identification process is shown in Figure 3.

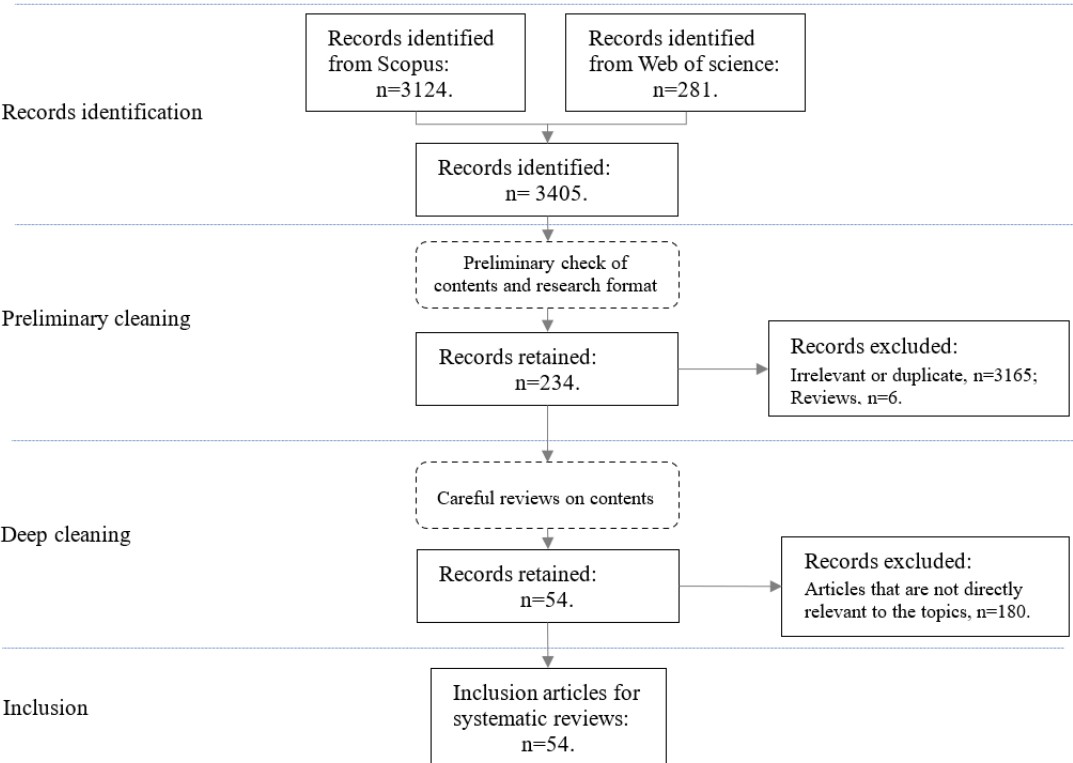

**Figure 3.** Process to identify the list of articles.

2.2.2. Statistics of the Identified Articles

There are several characteristics observed from the identified articles. Firstly, an analysis was conducted on the 14 journals where the identified articles were published. The majority of the identified articles were published in four journals: *Journal of Construction*

*Engineering and Management (20)*, *Journal of Management in Engineering (7)*, *International Journal of Project Management (4)*, *Sustainability (4)*, and *International Journal of Construction Management (4)*, which, in total, account for 72% of the total publications (as shown in Table 1).

**Table 1.** Volume of articles identified in the surveyed journals.

| The Surveyed Journals | Number of included Articles |
| --- | --- |
| Journal of Construction Engineering and Management | 20 |
| Journal of Management in Engineering | 7 |
| International Journal of Project Management | 4 |
| Sustainability | 4 |
| International Journal of Construction Management | 4 |
| Automation in Construction | 3 |
| Construction Management and Economics | 3 |
| Journal of Computing in Civil Engineering | 2 |
| Engineering, Construction and Architectural Management | 2 |
| Canadian Journal of Civil Engineering | 1 |
| Journal of Infrastructure Systems | 1 |
| Journal of Cleaner Production | 1 |
| Risk Analysis | 1 |
| Safety Science | 1 |

It can be observed that very few publications were published between 2001 and 2005, with a steady increase in publications afterwards. Figure 4 shows that between 2001 and 2005, only one publication was found; but in the periods between 2006 and 2010 and 2011 and 2015, the number of publications increased by 11 and 13 in each time period, respectively. Finally, the total number was found to be 54.

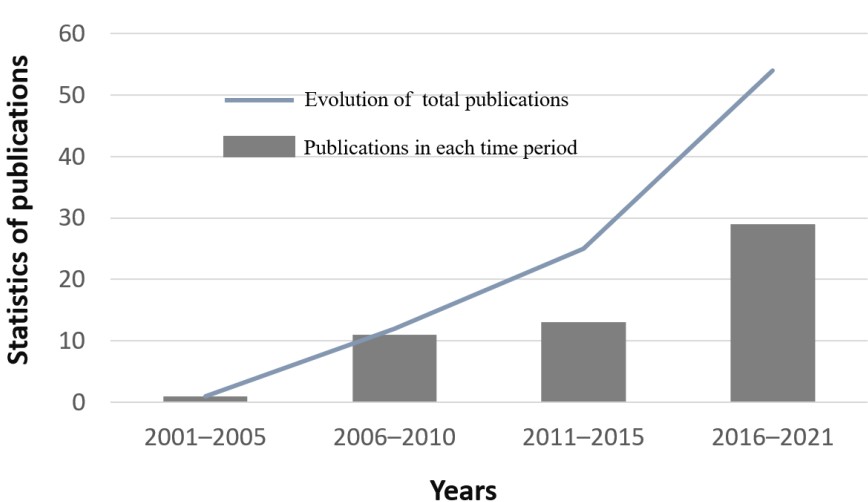

**Figure 4.** Volume of articles identified in each time period.

## 3. Research Content Analysis

### 3.1. Identifications of Research Contents

After conducting a content analysis on the identified papers, the authors found that the researches could be classified into three groups: (a) research on the influence of risk on cost performance, (b) research on the influence of risk on schedule performance, and (c) research on the influence of risk on multiple project goals. As shown in Figure 5, these three main contents accounted for 20 articles (37%), 16 articles (30%), and 18 articles (33%), respectively.

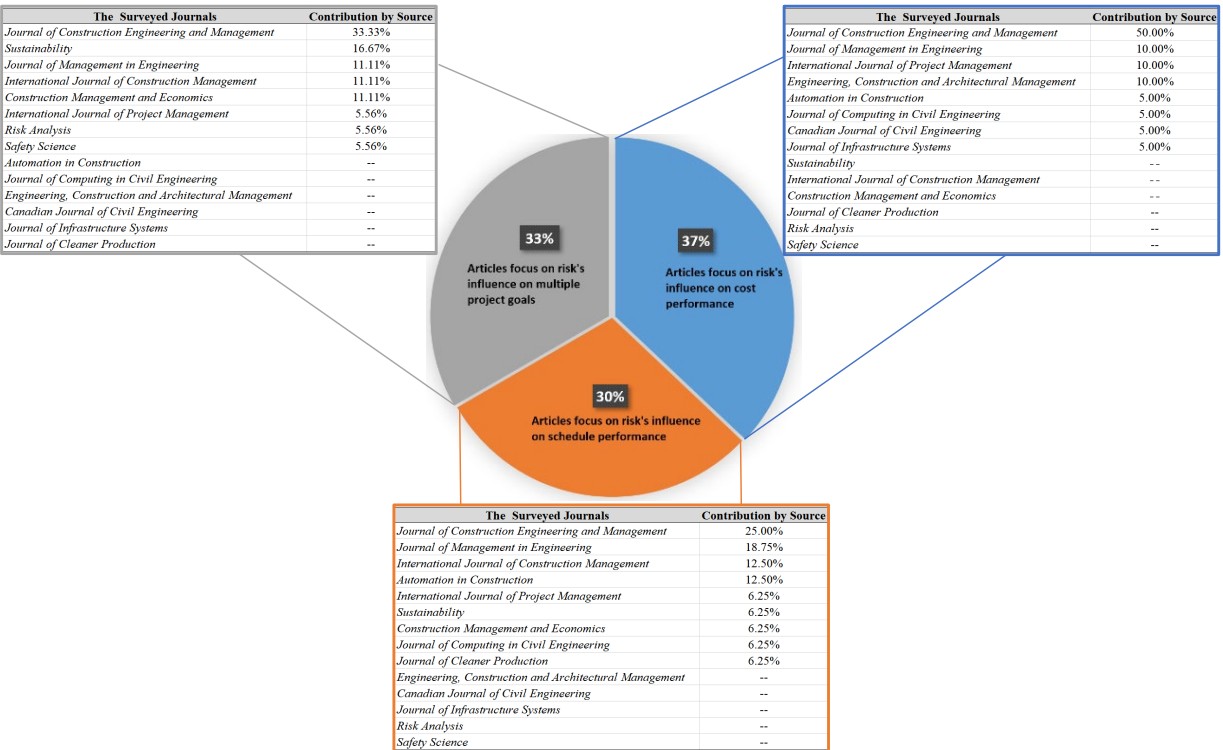

**Figure 5.** Research contents and the articles' source.

Furthermore, the contribution source of each article was also obtained for statistical purposes. The findings show that (i) the main contribution source of the research on the influence of risk on cost performance is the *Journal of Construction Engineering and Management*; (ii) the main contribution sources of the research on the influence of risk on schedule performance are the *Journal of Construction Engineering and Management* and the *Journal of Management in Engineering*; (iii) the main contribution sources of the research on the influence of risk on multiple project goals are the *Journal of Construction Engineering and Management* and *Sustainability* (As shown in Figure 5).

### 3.2. Analysis of the Research Content

#### 3.2.1. Research on the Influence of Risk on Project Cost Performance

Cost control is one of the most important operations in project management; however, cost overruns have become common in projects for decades [3]. This cost failure is due to the presence of complexity and risk in projects [5,11,32]. For this reason, many researches have concentrated on the influence of risk on project cost performance. According to the analysis in this paper, these studies can be classified into two groups. The first group mainly concentrates on developing tools or approaches to assess the influence of risk on cost performance. These researches include developing tools or approaches to model the cost escalation uncertainty [3]; estimate cost overrun risk rating [32]; assess cost overrun risk [33]; calculate the budgetary impact of increasing the required confidence level in a probabilistic risk assessment for portfolio projects [34]; assess risk-related variables that may lead to cost overrun [35,36]; and evaluate and quantify the potential differences in project cost attributable to the selections of the project-delivery method [37].

The second group mostly focuses on identifying elements or paths that have an influence on the cost performance. Khodakarami and Abdi (2014) and Afzal et al. (2020) discussed the interdependency of risk factors and the interdependencies' influence on cost performance [6,38]; Eybpoosh et al. (2011), Mathew et al. (2021), and Zhao et al. (2018) identified risk factors and the causal relationship paths that exist among these risk factors and their direct or indirect effect on cost performance [10,11,39]; and Jung and Han (2017)

examined the influence of risk management on cost performance [40]. Additionally, several studies in this area also focused on cost contingency. Sonmez et al. (2007) identified the risk factors impacting cost contingency [41], while Diab et al. (2017) tested the influence of the perceived ratings of the cost risk drivers and the relative importance of these risk drivers on cost contingency amounts [42].

Through the analysis above, it can be observed that prior researches focused mostly on two aspects in the area of cost performance: (1) cost: develop tools or approaches to evaluate risk of cost overruns and cost contingency, analyze the causes of cost overruns, and compare the cost overrun risk in different project delivery methods; and (2) identify risk elements, analyze the causal relationship paths among these elements, and discuss their impact on cost performance. However, several limitations are observed in these researches. The researches have been mainly limited to risk identifications and risk dependency analysis and evaluating their impacts on project performance. However, only in rare instances has the interaction process between the decision of the stakeholders and the feedback effect of project performance on the stakeholders been considered.

### 3.2.2. Research on the Influence of Risk on Project Schedule Performance

Construction project delays are among the most pressing challenges faced by the construction sector globally [43]. These delays are attributed to the sector's complexity and the inherent interdependency of risk factors [44]. Therefore, a large number of researches on the influence of risk on project schedule performance have been observed. A majority of these researches focused on developing tools or approaches to assess the influence of risk on schedule performance. Wiguna and Scott (2006) established a path model to analyze the impacts of risks and risk paths on project schedule performance [45]; Xu et al. (2018) developed a hybrid dynamic approach for investigating the effect of risks on project schedule performance [4]; Chen et al. (2021) developed a novel Bayesian Monte Carlo simulation-driven approach for construction schedule risk inference [46]; Gondia et al. (2020) developed machine learning models to facilitate accurate project delay risk analysis and prediction [44]; Balta et al. (2021) developed a Bayesian Belief Network based risk assessment method for delay prediction [47]; Schatteman et al. (2008) developed an integrated methodology for planning construction projects under uncertainty [48]; and, finally, Tokdemir et al. (2019) proposed a delay risk assessment method for project scheduling [49].

Other studies in this area investigated the causes of project delays. Russell et al. (2013) discussed what causes people to add and size time buffers [50]; Wambeke et al. (2013) used a risk assessment matrix to determine which causes of variation posed the greatest risk to schedule performance [51]; and Khalesi et al. (2020) tried to identify and reduce time delays caused by rework in construction projects [52]. Moreover, the interaction of risk factors and their influence on the schedule performance were also discussed. For instance, Luu et al. (2009) described how the Bayesian belief network is applied to quantify the probability of construction project delays [9]; Li et al. (2016) recognized and investigated schedule risks in the stakeholder-associated network of Hong Kong prefabrication housing construction projects [53]; and Wang and Yuan (2017) took a holistic view to investigate the overall dynamics of infrastructure project risks and how dynamic risk interactions can affect a project's schedule performance [7].

From the analysis above, it was found that prior researches in this area have mostly concentrated on developing tools or approaches to assess the influence of risk on project schedule performance. This includes developing tools or approaches to evaluate the influence of risk on project performance, conduct risk inference, predict project delays, and carry out project planning under uncertainty. Other researches focused on investigating the causes of project delays and discussing the interactions of risk factors and their influence on schedule performance. However, some limitations exist in these researches. For instance, these researches have rarely considered the decision of stakeholders on schedule risk or concentrated on the influence of project performance on project stakeholders.

### 3.2.3. Research on the Influence of Risk on Multiple Project Goals

Since it is not feasible for stakeholders to focus on a single project goal, project performance criteria need to include multiple objectives. Hence, research on this area should place more emphasis on the effects of risk on multiple objectives in a project. Thirty-three percent of the researches identified focused on multiple objectives. Most of them focused on developing tools or approaches to assess the influence of risk on multiple project objectives. Odeyinka et al. (2013) developed a cost flow approach-based model to assess the impacts of risk occurring on project performance [5]; Monzer et al. (2019) proposed an approach for construction risk assessment [2]; Assaad et al. (2020) created a holistic framework to quantify the impacts of the risks related to the performance of projects [16]; Su et al. (2021) established a systematic approach to evaluate the influence of risk on project performance [18]; Dikmen et al. (2012) developed a web-based tool to calculate risk-adjusted duration and cost [17]; and Farshchian and Heravi (2018) developed an agent-based simulation model to evaluate time and cost uncertainties in portfolio projects [54].

Other researches focused on the identification and evaluation of risks and their influence on multiple project goals. El-Sayegh and Mansour (2015) focused on project-performance-related risk identification [15]; Papanikolaou and Xenidis (2020) discussed how risks influence multiple project goals [55]; Mojtahedi et al. (2010), Al-sabah et al. (2014), and Liu et al. (2017) concentrated on risk identification and evaluation [19,56,57]; Hwang et al. (2014) investigated the influence of risk management on project performance [58]; and Rodríguez-Rivero et al. (2020) identified links between risk management and project success [59]. Moreover, some other researches also focused on investigating the major causes of poor time and cost performance [8], developing methodologies to select risk response actions [13] and discussing what enables high-risk projects to yield high returns [60].

Therefore, it can be observed that a wide variety of researches on the influence of risk on multiple project objectives have concentrated on developing tools or approaches for risk influence assessment. And researches that focused on investigating the causes of poor time and cost performance, developing methodologies to select risk response actions, and discussing what enables high-risk projects to yield high returns, have also been discussed. However, the limitations are also obvious. That is, the researches that focus on the decisions of stakeholders and the influence of project performance on stakeholders have rarely been considered in this group.

## 4. Research Method Analysis

### 4.1. Research Methods Adopted in the Identified Articles

Since the research method is vital for the accuracy of the research results, the selection of research method is also critical to address the influence of risk on project performance, and the research methods may also change over time depending on the level of technology development. Therefore, this paper also reviews the research methods that were applied in the identified articles to report on them.

### 4.1.1. Process to Identify the Research Method

Construction management is a relatively new field, which draws from both the natural and social sciences. As such, many different theories of knowledge or paradigms compete for methodological primacy [61]. According to Agyekum-Mensah et al. (2020), this phenomenon reflects the methodological pluralism in construction engineering and management research methods [62]. For instance, in some occasions, there exists a widespread confusion over terms such as 'method', 'methodology', and 'paradigm' in construction research methods [61]; and techniques such as triangulation, facilitation, complementarity, and multimethodology should be encouraged in construction engineering and management research [61,62]. Hence, it is a challenge to identify and classify all the research methods in the construction engineering and management domain. Thus, this paper will

focus only on the research methods in the identified articles and provide a discussion of each of them.

Moreover, given the pluralism of the research methods, as mentioned by Dainty (2008), the ones used in construction engineering and management cannot always be decomposed in classifications, and some research methods may also include other methods. For instance, in structural equation modeling (SEM), we usually need to conduct a literature review and questionnaire surveys before fitting the multi-equations. Hence, this paper introduces the concept of main research method (MRM). An MRM refers to the main research method that was applied in the identified article in this paper. In the SEM case above, SEM is the MRM, while the literature review and the questionnaire surveys are supporting methods used for the application of SEM. Finally, this paper applied the process and concepts proposed above to the analysis of the identified articles for identifying the research methods.

4.1.2. Analysis of Research Methods

a.      Research methods applied in each time period

Table 2 shows that the number of research methods that were adopted in each period has increased steadily over the years. In the period between 2001 and 2005, only one research method was adopted. That is, Jannadi and Almishari (2003) established a risk assessment model through an analysis of risk definition and used the sensitivity analysis method to assess the influence of risk on project performance [63]. However, in the period between 2006 and 2010, the number increased to 13. In this period, some risk management methods that were usually implemented in other industries were introduced into the construction engineering industry. For instance, Imbeah et al. (2009) used the case study method to introduce the Advanced Programmatic Risk Analysis and Management Model (APRAM), originally developed for the aerospace industry, into risk management in construction engineering [64]. Besides that, research methods such as structural equation modeling, distribution fitting, and fuzzy techniques, that were frequently found in the management science area, were also found to have been adopted in construction in this period. In the periods between 2011 and 2015 and 2016 and 2021, the research methods that were adopted increased to 20 and 23, respectively. Research methods including questionnaire surveys, structural equation modeling, expert interviews, discrete event simulation, Monte Carlo simulation, fuzzy techniques, system dynamics simulation, and index creation were found to have been widely implemented. For the total targeted period in this paper, between 2001 and 2021, the types of research methods found were 36, as shown in Table 2.

b.      Research methods applied in each research content

There are several characteristics that can be observed from analyzing the relationships between the research methods and the research contents in the identified papers, as shown in Figure 6. First of all, the size of the solid circles on the right side in Figure 6 shows the frequency of each method's application in the identified articles. That is, it can be concluded that questionnaire survey is the most frequently adopted method, whereas research methods such as stochastic processes and TOPSIS were not commonly observed, etc. Secondly, some research methods were frequently and likely used in some of the research contents. For instance, agent-based simulation and index creation were mostly used in the research on the influence of risk on schedule performance; social network analysis and machine learning were mostly adopted in the research on the influence of risk on multiple project goals; and regression analysis was mostly implemented in the research on the influence of risk on cost performance. Thirdly, there are also some research methods that are applicable to multiple research contents. For example, structural equation modeling, case analysis, questionnaire survey, system dynamics simulation, and expert interviews can be widely implemented in different types of research contents. These characteristics systematically depict the applications of the identified research methods in each research content.

**Table 2.** Research methods applied in each period.

| Research Methods | 2001–2005 | 2006–2010 | 2011–2015 | 2016–2021 | All (2001–2021) |
|---|---|---|---|---|---|
| Interaction analysis of networks [17] | – | – | 1 | – | 1 |
| Agent-based simulation [18] | – | – | – | 2 | 2 |
| Analytical network processing [38] | – | – | – | 1 | 1 |
| Principal component analysis [57] | – | – | 1 | – | 1 |
| Structural equation modeling [45] | – | 1 | 3 | 3 | 7 |
| Sensitivity analysis [34] | 1 | 1 | – | 3 | 5 |
| Social network analysis [51] | – | – | 1 | 1 | 2 |
| Step-Wise Weight Assessment Ratio Analysis [52] | – | – | – | 1 | 1 |
| Stochastic processes [54] | – | – | – | 1 | 1 |
| TOPSIS [56] | – | 1 | – | – | 1 |
| Comparative analysis (based on the Mann Whitney U test method) [60] | – | – | – | 1 | 1 |
| Case analysis [52] | – | 1 | 1 | 1 | 3 |
| Bayesian techniques [9,33] | – | 2 | 1 | 2 | 5 |
| Distribution Fitting [3] | – | 1 | – | 2 | 3 |
| Risk matrix analysis [40] | – | – | 1 | 1 | 2 |
| Regression analyses [41] | – | 1 | 1 | 2 | 4 |
| Machine learning [44] | – | – | – | 2 | 2 |
| Risk structuring based on WBS [13] | – | 2 | – | – | 2 |
| Discrete event simulation [37,48] | – | 1 | 2 | 2 | 5 |
| Monte Carlo simulation [17] | – | – | 2 | 2 | 4 |
| Fuzzy techniques [32] | – | 2 | – | 4 | 6 |
| Line-of-balance (LOB) [49] | – | – | – | 1 | 1 |
| Artificial neural network [5] | – | – | 1 | – | 1 |
| Data-mining techniques [37] | – | – | 1 | – | 1 |
| Numerical simulation [3] | – | 1 | – | – | 1 |
| Questionnaire survey [50] | – | – | 4 | 6 | 10 |
| System dynamics simulation (SDS) [4] | – | – | – | 4 | 4 |
| Systematic modeling [17] | – | – | 1 | – | 1 |
| Coefficient analysis [37] | – | 1 | 1 | – | 2 |
| Workshop [37] | – | – | 1 | – | 1 |
| Histogram data analysis [50] | – | – | 1 | – | 1 |
| Index creation [57] | – | – | 2 | 2 | 4 |
| Expert interviews [45] | – | 1 | – | 5 | 6 |
| Ontology [65] | – | – | 1 | – | 1 |
| Cyclic operation network [66] | – | – | 1 | – | 1 |
| Interpretive Structural Modeling [67] | – | – | – | 1 | 1 |
| Types of research methods | 1 | 13 | 20 | 23 | 36 |

### 4.2. Applications of the Most Frequent Research Methods

Figure 6 shows all the research methods that were frequently used in the identified articles. However, the number of research methods that were implemented more than three times was found to be 13. In this section, a deep understanding is provided on the applications of these 13 frequently used research methods, and a systematic review on each one of them was conducted.

a.    Structural Equation Modeling

Structural Equation Modeling (SEM) is a mathematical and deterministic approach derived from multivariate statistical analysis and can be used to analyze multiple inter-relationships among variables and integrate the impact of unobserved relationships that may occur among them [11]. In the identified articles, SEM was used to examine the effect of risk factors on project performance [11,19,45]; identify and assess causal relationships among risk factors and their effect on cost overrun [10]; predict cost overrun [35]; and assess the paths of risks associated with project performance [35,39]. This shows that SEM is effective in modeling the interdependency between risk factors and their effects on project performance.

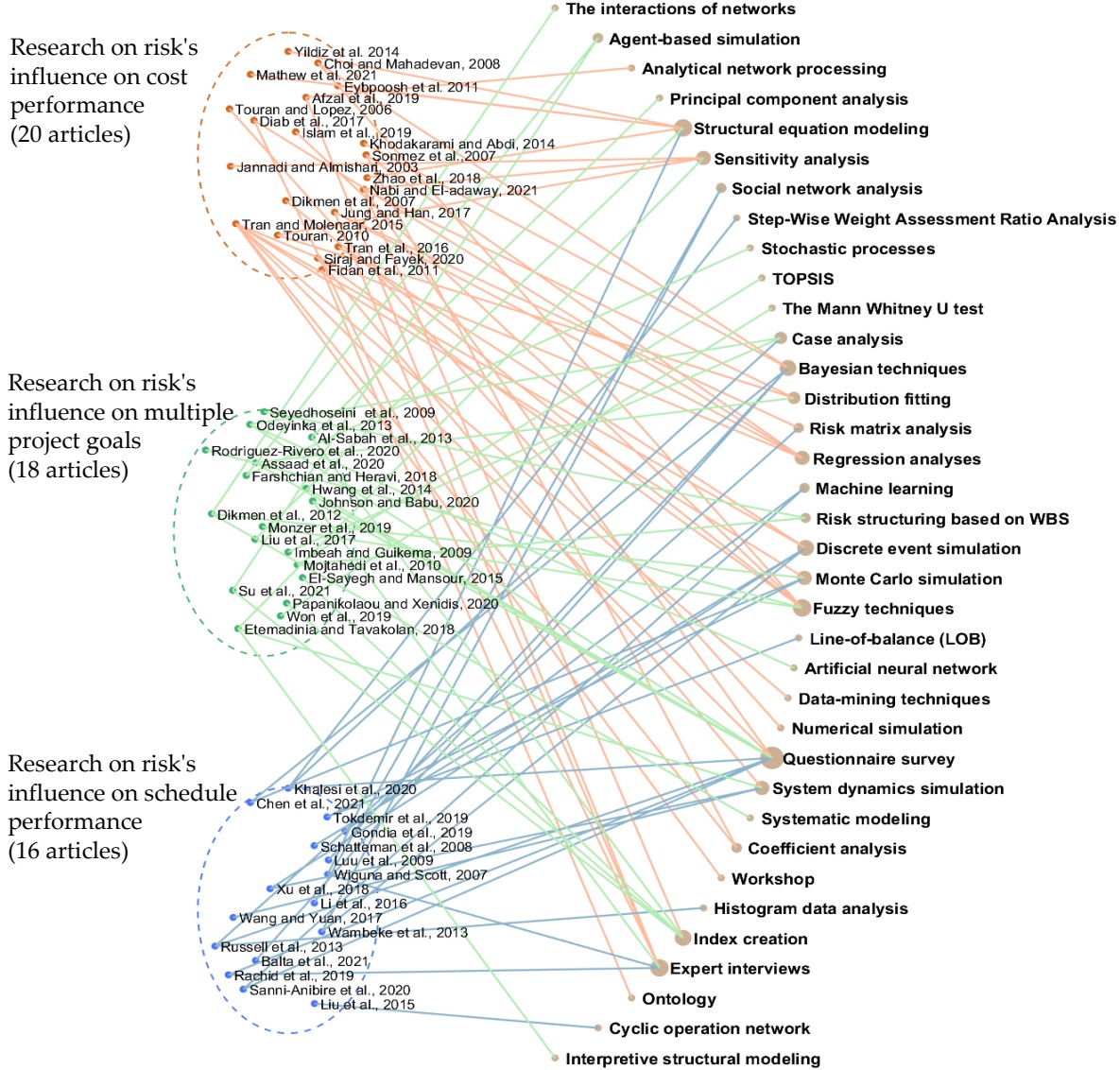

**Figure 6.** Relationship between research method and research content. Research on risk's influence on cost performance [3,6,10,11,27,32–42,63,65,68,69]. Research on risk's influence on multiple project goals [4,7,9,19,44–53,69,70]. Research on risk's influence on schedule performance [2,5,8,13,15–19,54–60,64,67].

b. Bayesian Techniques

In this paper, Bayesian techniques mainly refer to Bayesian networks and the Bayesian updating technique. Bayesian networks are a framework that presents probabilistic relationships and enables probabilistic inferences among a set of variables [6]. The Bayesian updating technique is an approach to combine subjective data and observed data in a manner that considerably improves the quality of the subjective input data [71]. Five of the identified articles have adopted Bayesian techniques in their research. In these researches, Bayesian networks were used to quantify the probability of construction project delays based on cause–effect relationships analysis [9], analyze the dependencies of cost items [6], conduct schedule risk inferences [46], and develop a decision-support tool for delay prediction through modeling the interrelations between risk factors [47]. Finally, Choi and Mahadevan (2008) also used a fuzzy Bayesian updating technique to identify and evaluate the critical risk events in construction projects [33]. Therefore, in the prior researches, Bayesian techniques were mostly adopted to analyze and model risk networks.

c.    Distribution Fitting

Distribution fitting is an approach used to select the statistical distribution that best fits the data. Distribution fitting can be classified into two groups, namely, parametric distribution fitting and nonparametric distribution fitting. Parametric distribution fitting consists in fitting the data to known theoretical distributions such as the normal, extreme value, uniform, and triangular distributions [36]. On the other hand, nonparametric distribution fitting refers to functions whose distributions do not have predefined parameters. In this case, polynomial function fittings are commonly used [16,36].

In the researches identified, distribution fitting was found to be applied in four articles, and these applications can be classified into two groups. In the first group, the distribution fitting methods were separately used. That is, Touran and Lopez (2006) used the parametric distribution fitting to model individual project costs in the market [3]; Schatteman et al. (2008) used the parametric distribution fitting to relate the impact of risk on duration probability [48]. In the second group, parametric distribution fitting and nonparametric distribution fitting were combined together as a hybrid method. For example, Assaad et al. (2020) used the hybrid method to conduct the distribution fitting of risk impacts [16], and Nabi and El-Adaway (2021) used the hybrid method to fit the impact distributions of each modular risk [36]. Hence, distribution fitting techniques are a useful tool in evaluating the risk impact in systematic simulations, and in some occasions, parametric distribution fitting and nonparametric distribution fitting can be implemented together.

d.    Regression Analysis Techniques

Regression analysis is a well-known and commonly used statistical learning technique used to infer the relationship between a dependent variable and independent variables. In in this paper, regression analysis techniques refer to linear regression analysis, multi-coefficients regression analysis, stepwise regression analysis, and hierarchical regression analysis. In prior studies, multi-coefficient regression analysis techniques were used in quantifying the impacts of risk factors on contingency [41] and investigating the interaction between delivery risk factor characteristics and project-delivery method selection [37]. Stepwise regression analysis was adopted to analyze the dependency relationship between predetermined cost contingency amounts and the perceived ratings of risk drivers [42]; hierarchical regression analysis was used to examine the impacts of the risk management process on cost performance [40]. These researches show the application of regression analysis techniques in the area of risk influence.

e.    Discrete event simulation

Discrete event simulation (DES) is used to simulate the impact of discrete events on the simulated systems. Four of the identified articles were found to have adopted discrete event simulation in their studies. These researches used DES techniques to analyze the impact of risk disruptions on project scheduling [48], studying how task start time and duration variations affect the whole management system of project [51], and developing a hybrid dynamic approach to investigate the effect of risks on project schedule performance [4]. The DES technique was also used in cross-impact analysis, which is an analytical approach to determine the overall effect on the probability of a variable based on chains of impact from related variables. For instance, the DES technique was used to capture the uncertainties and interactions among the input and decision variables [37]. Through these studies, it can be concluded that discrete event simulation techniques are suitable for understanding the impacts of risk disruptions on project risk management systems.

f.    Monte Carlo Simulation

Monte Carlo simulation is a probability-based technique in which a parameter is modeled by a probability distribution and the results are calculated multiple times, each time using a different value randomly selected from the defined distribution [49]. Five of the identified studies were found to have used Monte Carlo simulation for various

reasons. Monte Carlo simulations were used to create scenarios for simulating the cost–duration interinfluence [17]; modeling the varied probability of delivery-risk factors and capturing the probabilistic information regarding the propagation of risk and uncertainty in a project [37]; producing probability distributions for possible outcome values for delay risk scenarios [49]; and creating risk data for Bayesian inference [46]. In these studies, Monte Carlo simulation was adopted when there was a lack of data for simulation or risk inference. Hence, Monte Carlo simulation was useful for scenario creation in the context of risk analysis.

g.  Fuzzy Techniques

In this paper, fuzzy techniques mainly refer to fuzzy logic and fuzzy sets. Fuzzy logic is a logic that is used to describe fuzziness, with certain degrees used as representation (e.g., 0~1) to describe all things. It is different from traditional logic, that represents things as true (1) and false (0), because fuzzy logic can represent things that are partially true (e.g., 0.7) or partially false (e.g., 0.2). In recent years, this approach has been applied to several areas, such as the identification of stakeholders, transportation, and energy [59]. Fuzzy sets are mathematical sets based on the theory of fuzzy logic and can be used to calibrate vagueness. In the identified articles, fuzzy techniques were used to capture experts' knowledge and rate cost overrun risks in projects [32]; handle ambiguity and model subjective probability judgment [33]; elicit experts' judgments for assessing cost overrun risks [72]; obtain experts' opinions in heterogeneous groups for analyzing the probabilities and impacts of project risks [2]; and assess project team members' opinions and identify the links between risk management and project success [59].

h.  Questionnaire survey

A questionnaire survey is a common method used to elicit responses from experts regarding professional or experiential information. This information then undergoes processing for the data collected, along with aggregating and analyzing it to report on the findings. In the identified articles, questionnaire surveys were often used to study the causes of project performance variations. More specifically, these applications include identifying the most frequent and severe reasons for adding time buffers to construction task durations [50], analyzing the major causes of poor time and cost performance [8], and weighing and ranking the causes of rework in construction projects [52]. Questionnaire surveys were also adopted to determine the significant risk factors impacting the forecasted cost flow [5] and how risk factors impact the selection of project delivery methods [37]; investigate the impact of risk management on project performance [58]; and study the effects of risk factors on project performance [16,36].

i.  Index creation

Index creation is an important technique to assess or evaluate a research target. The technique was found to be implemented in four studies in the identified articles. Al-Sabah et al. (2014) established and used a relative importance index and a significance score to evaluate the impact of construction risks [57]; El-Sayegh and Mansour (2015) used a relative importance index to evaluate risk priority [15]; Papanikolaou and Xenidis (2020) established a series of indices to assess the risk-informed performance of construction projects [55]; and Nabi and El-adaway (2021) established an index to quantify modular risks for simulations [36]. In these researches, indices were established as a tool or evaluation system to assess the characteristics of certain research areas.

j.  Expert Interviews

Expert interview techniques are used to elicit experts' professional or experimental information. They are different from questionnaire surveys since they involve an interaction between an interviewer and interviewee. Interviews can be structured or semi-structured, depending on the format used. Structured interviews usually follow a rigid questioning process and order, whereas semi-structured interviews allow room for changes depending on the experts' response during the interview. In the identified articles, this technique was

mainly used in investigating the relationship between risk and project performance [45], evaluating the influence of the risk drivers on the amount of cost contingency available [42], exploring the critical elements of complexity–risk interdependency for cost–chaos in the construction management domain [38], and analyzing the major causes of poor time and cost performance [8]. These researches implied that expert interview techniques can be implemented in a wide range of occasions and are helpful in understanding the influence of the relations between two observed variables through eliciting experts' knowledge.

k. System Dynamics Simulation

System dynamics simulation (SDS), an approach based on cause–effect relationships, is a viable option for modeling and analyzing construction risks that addresses certain limitations of traditional risk analysis techniques [73]. Xu et al. (2018) used the SDS approach to describe and simulate the causal loop and interaction relations among project tasks/activities in schedule risk prediction simulations [4]; Wang and Yuan (2017) adopted the SDS approach to describe and investigate the interaction effects of the causal loop relations on project schedule delays [7]; Siraj and Fayek (2021) used the SDS approach to define the dynamic causal relationships among risk and opportunity events and quantify their impact on work package and project contingencies [73]; Etemadinia and Tavakolan (2021) adopted an SDS approach to analyze the uncertainties in the design phase of the construction projects [67].

It can be observed that SDS is capable of modeling the inter-relations and interactions of risk factors through a holistic view of the whole project; and quantifying the effects of these inter-relations and interactions on project performance. Moreover, through analyzing the methodologies applied in researches above, it can also be found that SDS can be coupled with other research methods such as discrete event simulation or fuzzy techniques.

l. Sensitivity Analysis

Sensitivity analysis is an approach used to indicate the accuracy and reliability of a model's output to variations in the input variables [72]. In risk management, this can aid decision makers in identifying the critical risk factors that are most capable of influencing project duration [49]. Jannadi and Almishari (2003) established a risk assessment model through an analysis of risk definition, and then used sensitivity analysis to examine how the severity, exposure, and probability of risk factors affect project performance in the risk assessor model [63]; Touran (2010) used sensitivity analysis to evaluate how variations of the probability of sufficiency of the portfolio budget affect the probability of sufficiency of each project's budget [34]; Etemadinia and Tavakolan (2021) used sensitivity analysis to assess the influence of risk on project performance and then prioritized and identified the major risks [67]; Nabi and El-adaway (2021) used sensitivity analysis to study how the changes of modular risks affect the cost performance of modularization in construction projects [36]. Additionally, sensitivity analysis has been used in the verification of models. For instance, sensitivity analysis was used to test the model outcome [49] and examine the sensitivity of the proposed risk assessment models to individual risk factors [72]. These studies show that sensitivity analysis can be used to examine how the changes in input variables affect the output in a risk assessment model. It was mainly used in areas such as critical risk identification, risks prioritization, and model verification.

m. Case Analysis

Case analysis is an approach that can be used to study complex phenomena and develop new theories. According to Tasci et al. (2020), the case study method should have its rules and standards, such as mixed methods, theory building, or theory testing [74]. Another variation is an 'example study', which is similar to a case study but does not conform to these rules and standards. Previous researches have not shown a clear distinction between these two types; hence, this paper will use the umbrella of 'case analysis' to depict both types. Several studies have adopted case analysis in their methodology. These applications include examining the effectiveness of a novel approach [52]; collecting data [5];

and demonstrating the applicability of an approach that originates from another field [64]. Additionally, case analysis is also useful in validating or testing novel theories, tools, or approaches. Examples include validating a risk quantification approach [9]; demonstrating the functions and performance of a risk management tool [35]; and verifying the risk inference approach [46]. These studies show that the case analysis method is applicable in examining or demonstrating the effectiveness of novel approaches in risk management as well as validating and testing novel theories and approaches.

## 5. Discussion

### 5.1. Status of Current Research

Knowledge of the influence of risk on project performance is important to risk management in construction engineering and management. With this knowledge, project stakeholders would be able to understand the mechanism of risk impact, conduct accurate project performance estimations, select suitable risk response actions, and finally improve project performance universally. From the review conducted on the identified studies, it can be observed that the research on the influence of risk on project performance has emerged in the past two decades. These researches can be classified into three groups, namely, the influence of risk on cost performance, the influence of risk on schedule performance, and the influence of risk on multiple project goals. Researches that focused on risk factor identification or risk interdependency modeling were the most common [6,7,9–11,15,19,38–40,53,55–59]. Other researches also investigated the cause of poor project performance [8,50–52]; evaluated the risk's impact on cost contingency [41,42]; discussed the risk response actions [13]; or discussed what enables high-risk projects to yield high returns [60]. These researches contribute to the body of knowledge regarding the influence of risk on project performance to date.

Regarding the research methodology, 36 research methods were found to be used by the 54 identified articles. The characteristics of the applications of each research method can be observed by analyzing the relationships between research methods and research contents. Moreover, research methods such as agent-based modeling and simulation, Bayesian techniques, distribution fitting, and machine learning were also implemented in previous studies (as shown in Table 2). By investigating the applications of the 13 most frequent research methods, it was found that (a) structural equation modeling was used to model the interdependency of risk factors and their effects; (b) Bayesian techniques were mostly adopted to analyze and model risk networks; (c) distribution fitting techniques were used to evaluate risk impacts in systematic simulations; (d) regression analysis was used to infer the relationship between a dependent variable and independent variables; (e) discrete event simulation was used to model the impacts of risk disruptions on project risk management system; (f) Monte Carlo simulation was a helpful tool in scenario creation; (g) fuzzy techniques were usually adopted to capture experts' knowledge or judgments; (h) questionnaire survey techniques were often used in studying the causes of project performance variations; (i) index establishing and analysis was used as a tool or evaluation system to assess the characteristics of research targets; (j) expert interviews can be widely adopted in different research topics; (k) system dynamics simulation is capable of modeling and quantifying the inter-relations and interactions of risk factors, and their effects on project performance; (l) sensitivity analysis can be used in critical risk identifications, risks priority, and model verification; and (m) case study is applicable in examining the effectiveness of novel approaches.

### 5.2. Gaps in the Identified Researches

Although a large number of studies have concentrated on the influence of risk on project performance, some gaps exist in the literature. Firstly, the accuracy in the quantitative researches can be improved. As shown in Table 3, 21 researches have focused on developing tools or approaches to assess the influence of risk on project performance. However, researches that evaluate the exact impacts of risk on cost or schedule variations

using real data have not been widely seen. Secondly, according to the analysis above, novel research methodologies that can be used in conducting more accurate assessments also need to be developed. For instance, building information modeling (BIM) can be combined with simulation techniques to conduct real-time risk influence simulations on an asset. Thirdly, it was found that few studies have taken into consideration the effect of participants' decision-making on the risks. That is, these researches have focused on the impacts of risk factors and their interactions but overlooked the influence of project participants' real-time behaviors in project management. Fourthly, very few researches have studied the entire interaction process (as shown in Figure 1) as a system to conduct the risk influence research.

**Table 3.** The main research methods (MRMs) adopted in tools or approaches.

| Authors | Tools or Approaches | The Main Research Methods Adopted |
| --- | --- | --- |
| Monzer et al. 2019 [2] | "an approach" | Fuzzy Techniques |
| Touran and Lopez, 2006 [3] | "a system" | Numerical simulation |
| Xu et al., 2018 [4] | "a hybrid dynamic approach" | System Dynamics Simulation; Discrete Event Simulation |
| Odeyinka et al., 2013 [5] | "a cost flow approach-based model" | Questionnaire survey; Case Analysis; Artificial neural network |
| Assaad et al., 2020 [16] | "a holistic framework" | Questionnaire survey; Distribution Fitting |
| Dikmen et al., 2012 [17] | "a Web-based tool" | Interaction analysis of networks; Monte Carlo Simulation; Systematic Modeling |
| Su et al., 2021 [18] | "a systematic approach" | Agent-based simulation |
| Dikmen et al., 2007 [32] | "a fuzzy risk assessment approach" | Fuzzy Techniques |
| Choi and Mahadevan, 2008 [33] | "a risk assessment methodology" | Fuzzy Techniques; Bayesian Techniques |
| Touran, 2010 [34] | "a mathematical model" | Distribution Fitting; Sensitivity Analysis |
| Yildiz et al., 2014 [35] | "a knowledge-based risk mapping tool" | Structural Equation Modeling |
| Nabi and El-adaway, 2021 [36] | "a risk-based approach" | Questionnaire survey; Index creation; Distribution Fitting |
| Tran and Molenaar, 2015 [37] | "a risk-based modeling methodology" | Questionnaire survey; Workshop; Discrete Event Simulation; Regression Analyses; Data-mining techniques; Coefficient analysis; Monte Carlo Simulation |
| Gondia et al., 2020 [44] | "machine learning models" | Machine Learning |
| Wiguna and Scott, 2006 [45] | "a path model" | Expert interviews; Structural Equation Modeling |
| Chen et al., 2021 [46] | "a novel Bayesian Monte Carlo simulation–driven approach" | Bayesian Techniques; Monte Carlo Simulation |
| Balta et al., 2021 [47] | "a Bayesian Belief Network based risk assessment method" | Bayesian Techniques |
| Schatteman et al., 2008 [48] | "an integrated methodology" | Discrete Event Simulation |
| Tokdemir et al., 2019 [49] | "a delay risk assessment method" | Line-of-balance (LOB); Monte Carlo Simulation |
| Farshchian and Heravi, 2018 [54] | "an agent-based simulation model" | Stochastic processes; Agent-based simulation |
| Islam et al., 2019 [72] | "a modified fuzzy group decision-making approach" | Fuzzy Techniques |

## 6. Conclusions

Knowledge of the influence of risk on project performance is a vital part of risk management. For this reason, a systematic review of the research on the influence of risk on project performance has been carried out in this paper. Fifty-four articles were identified and classified into three groups according to their research contents. The research contents in each article and the research methods were reviewed, and the 13 most frequent research methods were discussed. It was found that many of the prior researches developed tools or approaches to assess the influence of risk on project performance, while other researches focused on the identification of risk factors or risk interdependency modeling, investigated the cause of poor project performance, evaluated risk impact on cost contingency, discussed the risk response actions, or discussed what enables high-risk projects to yield high returns. However, four gaps were identified from these researches, namely: a need to improve the

accuracy in quantitative research of the influence of risk on project performance; a need for novel research methodologies for conducting more accurate risk influence assessments; taking into consideration project participants' decision-making in their researches; and creating a framework that treats the risk influence assessment as a whole system. Since this research only focused on two project objectives (cost and schedule), recommendations for future research include expanding the focus to more project objectives.

**Author Contributions:** Conceptualization, G.S. and R.K.; methodology, G.S. and R.K.; validation, G.S.; formal analysis, G.S.; investigation, G.S.; data curation, G.S.; writing—original draft preparation, G.S. and R.K.; writing—review and editing, R.K. and G.S.; visualization R.K. and G.S.; supervision, G.S.; project administration, G.S. All authors have read and agreed to the published version of the manuscript.

**Funding:** This research was supported by Guilin University of Technology Foundation (RD2100002104) and Guangxi Science Foundation (2018GXNSFAA050145).

**Acknowledgments:** The authors acknowledge the contributions of Hastak M., from Purdue University, for the paper 'Risk Sharing Strategies for IPD Projects: Interactional Analysis of Participants' Decision-Making. Journal of Management in Engineering. 2021, 37(1), 04020101'.

**Conflicts of Interest:** The authors declare no conflict of interest.

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
