# Peer review of "Research on the Influence of Risk on Construction Project Performance: A Systematic Review"

_sustainability, doi:10.3390/su14116412_

Round 1
Reviewer 1 Report
In the paper a systematic review of the research on the influence of risk on project performance is conducted. The paper is well written and deals with an interesting topic. However the paper has some shortcoming in its current format. The Authors need to address the following comments and suggestions:
- Authors should unify the way of referring to literature. E.g. “… in the UAE construction industry [8] while Dikmen et al. (2012) used a…”
- What do the Authors mean by “the journals that have high reputation in the construction engineering and management area”? . Are these journals with a high IF?
- “... Siraj et al. (2019)’ s method for journal selections...” – Please briefly describe how this method works.
- The Authors write: „...the 14 journals identified by Siraj et al. (2019) were used to conduct the article survey...” and next “... Additionally, another four journals with a large number of publications in the construction engineering and management area or risk management and with high reputations in academic were also surveyed...”. So there should be 18 journals. But in part 3.2.2. the Authors write: “... Firstly, an analysis was conducted on the 14 journals where the identified articles were published…”. How many journals were considered then?
- I am afraid that Fig 6 will be illegible.
Reviewer 2 Report
This manuscript proposes a framework to depict the full scope of the research on risk management. Overall, this is a clear, concise, and well-written review report. However, the introduction part is not detail. The author mentioned that “no previous reviews have been conducted to report on the influence of risk on project performance”. In my eyes, this statement is not objective.
Author Response
Response to Reviewer 2 Comments
Dear reviewer,
Thank you very much for your insightful comments and careful review on our paper! Your comments are very valuable and helpful for improving the quality of our paper. Here are the responses to your comments respectively.
Point 1: The introduction part is not detail.
Response 1: We have restructured introduction. We hope this time it can meet your requirements. (in red)
Point 2: The author mentioned that “no previous reviews have been conducted to report on the influence of risk on project performance”. In my eyes, this statement is not objective.
Response 2: There are some reviews that have been concentrated on the risk in construction project; but this paper is different, the review focusses on the research on the influence of risk on project performance.
In the last but one paragraph of “1. Introduction”, the paper also elaborates the reasons. That is, “Previously, several reviews have been conducted on risk management. Khallaf et al. (2018) proposed a three-tier classification of risks in public-private partnership projects[12]; Taroun et al. (2011) and Taroun (2014) reviewed construction risk modelling and assessment[23,24]; Zou et al. (2017) concentrated on the review of risk management through BIM and BIM-related technologies[25]; Xia et al. (2018) conducted a review on the integrations of construction risk management and stakeholder management[26]; Siraj et al. (2019) reviewed the risk identification and common risks in construction[27]; Hegde and Rokseth (2020) carried out a review on the applications of machine learning methods in engineering risk assessment[28]. Many of these researches have focused either on risk identification or risk modeling and assessment. However, there have been no previous reviews that targeted the four groups as mentioned earlier.”
Besides that, to organize the sentence in a more accurately way, the paper would like to use anther sentence: “rare reviews have been conducted to fully report on the research on the influence of risk on project performance” to replace the previous one. (in red)

Reviewer 3 Report
The article presented is interesting and relevant.
In spite of the title, it is not probabilistic-mathematical, but sociological-philological and exploratory. There are, however, significant comments on its structure and presentation.
1. From the title it is not clear what risks the article is devoted to. There are a multitude of risks. The title should be more precise.
2. The abstract. Again - it is not clear what is the aim of the research. The authors write that previous studies have not described the impact of risk on project performance. Of course, it was studied and given in the introduction to the articles that the authors have in the literature. Another thing, the risks were not graded and assessed the weight of each type of risk, etc. The authors should clearly define the purpose of their work and the objectives they set for themselves in their research.
In the article review does not need to write exactly how many articles studied (54 pcs.), it is clear from the list of references. By the way, 54 is not so much, usually it is 80 to 100 for the review.
No need to write that you analyzed "content, methods, etc." - that's obvious. But the classification into three groups - what?, by what parameters, etc. Further, the 13 methods identified - which ones, what are their commonalities and differences? "Conclusions were drawn and recommendations were offered." Which ones? What did the authors "showed, found, established"? Everything written in the conclusion should be visible in the Abstract.
The abstract should be redone.
The purpose of the article and (and its title) is not clearly identified
3. Key words should be added: "construction", " journals in construction".
4. The introduction at the end of the section should more clearly indicate how previous reviews differ from this one and what the purpose and objectives of this article are.
Overall, the article is ready as an overview, it should be made more structured. Redo the abstract and strengthen the objectives of the paper in the introduction.
Author Response
Response to Reviewer 3 Comments
Dear reviewer,
Thank you very much for your careful review and insightful comments! Your comments are very valuable and helpful for improving the quality of our paper. Here are the responses to your comments respectively. We hope that the this revision can meet your requirments.
Point 1: Key words should be added: "construction", " journals in construction".
Response 1: Yes, the paper needs to convey its aim precisely. The authors would like to replace keyword “project performance” with “construction project performance”; and also change title of paper to “Research on the Influence of Risk on Construction Project Performance: A Systematic Review”. (in red)
Point 2: From the title it is not clear what risks the article is devoted to. There are a multitude of risks. The title should be more precise.
Response 2: The meaning that the title want to convey is “A systematic review on the research on the influence of risk on construction project performance”. It is an article of Review and this sentence is pretty long.
Under the umbrella of this title, all risks in construction project should be discussed (as it does in the paper). But as mentioned in the last sentence in “6. Conclusions” section, that is, “Since this research only focused on two project objectives (cost and schedule), recommendations for future research include expanding the focus to more project objectives”, the paper still has some limitations. That is, the paper only focusses on the paramount importance objectives (i.e., cost and schedule overruns [16]) in the project controls area, and has not been able to discuss the other objectives at the moment; and future reviews on this topic should expand the focus to more project objectives.
So, from that point of view, it seems the title of the paper still can make sense of the research to some extent. Thus, the authors would like to keep it unchanged. (in red)
Point 3: The abstract. Again - it is not clear what is the aim of the research. The authors write that previous studies have not described the impact of risk on project performance. Of course, it was studied and given in the introduction to the articles that the authors have in the literature. Another thing, the risks were not graded and assessed the weight of each type of risk, etc. The authors should clearly define the purpose of their work and the objectives they set for themselves in their research.
In the article review does not need to write exactly how many articles studied (54 pcs.), it is clear from the list of references. By the way, 54 is not so much, usually it is 80 to 100 for the review.
No need to write that you analyzed "content, methods, etc." - that's obvious. But the classification into three groups - what?, by what parameters, etc. Further, the 13 methods identified - which ones, what are their commonalities and differences? "Conclusions were drawn and recommendations were offered." Which ones? What did the authors "showed, found, established"? Everything written in the conclusion should be visible in the Abstract.
The abstract should be redone.
The purpose of the article and (and its title) is not clearly identified.
Response 3: Thank you very much for your kindly advice! Here is the revised abstract:
Knowledge on the influence of risk on project performance is an important part of risk management. Previous studies have concentrated on this area to identify risks in aspects of project performance such as cost and schedule. However, rare reviews have been conducted to fully report on the research on the influence of risk on project performance. For this reason, to identify and analyze such researches in these areas a systematic review was conducted in this paper. More specifically, in this paper, 54 relevant articles were identified and classified into three groups according to their research contents; the research contents in each article and the research methods were reviewed and the 13 most frequent research methods were also identified and discussed. It was found that most of the previous researches concentrated on developing tools or approaches to assess the influence of risk on project performance. Additionally, researches focused on risk factors’ identification or risk interdependency modeling were also common, along with researches that investigated the cause of poor project performance, evaluated risk impact on cost contingency, discussed the risk response actions, and discussed what enables high-risk projects to yield high return. However, four gaps were identified from these researches, which are: a need for improving the accuracy in quantitative research of the influence of risk on project performance; a need for novel research methodologies for conducting more accurate risk influence assessments; taking into consideration project participants’ decision-making in their researches; and creating a framework that treats the risk influence assessment as a whole system. Besides that, since this research only focused on two project objectives (cost and schedule), recommendations for future research include expanding the focus to more project objectives.
For the issue of the number of the identified articles, the authors agree that a number of 54 is not large for a review. In future, after expanding the focus to more project objectives, this number can be improved. But with a number of 54, it may be as well to conduct a review in this paper. As we have been known that, there are some other researches (for instance, Gómez-Marí’s article: https://www.mdpi.com/2071-1050/13/9/5097) even have a smaller number of identified articles in their review; besides that, this paper has also gone deep into these articles’ research contents and research methods, and made a systemic discussion. So, we wouldn't want to change the number at the moment.
For the issue of the purpose of the article(and its title), we hope that it has been expressed clearly in the revised introduction and abstract. (in red)
Point 4: The introduction at the end of the section should more clearly indicate how previous reviews differ from this one and what the purpose and objectives of this article are..
Response 4: Thank you very much! To solve this issue, the introduction section has been restructured.
I hope this time it can answer these queries. For instance, the last but one paragraph shows how previous reviews differ from this one and what the purpose and objectives of this article are. That is, “Previously, several reviews have been conducted on risk management. Khallaf et al. (2018) proposed a three-tier classification of risks in public-private partnership projects[12]; Taroun et al. (2011) and Taroun (2014) reviewed construction risk modelling and assessment[23,24]; Zou et al. (2017) concentrated on the review of risk management through BIM and BIM-related technologies[25]; Xia et al. (2018) conducted a review on the integrations of construction risk management and stakeholder management[26]; Siraj et al. (2019) reviewed the risk identification and common risks in construction[27]; Hegde and Rokseth (2020) carried out a review on the applications of machine learning methods in engineering risk assessment[28]. Many of these researches have focused either on risk identification or risk modeling and assessment. However, there have been no previous reviews that targeted the four groups as mentioned earlier. For this reason, a systematic review is needed on the influence of risk on project performance.” (in red)
